# VGGT-X: When VGGT Meets Dense Novel View Synthesis

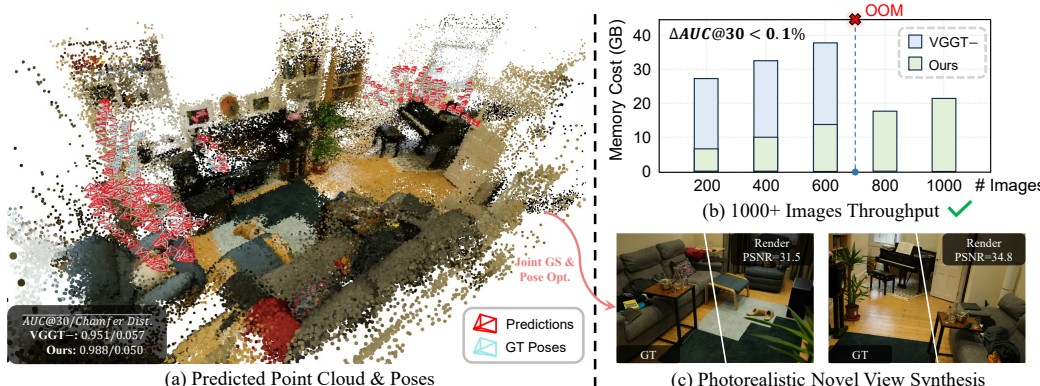

Figure 1: Reconstruction and Novel View Synthesis results. In part (a), we extend VGGT to handle dense multi-view inputs and incorporate an efficient global alignment, yielding highly accurate predictions. Part (b) demonstrates that eliminating redundant VRAM usage enables inference throughput over 1000 images without compromising performance. The VGGT− here denotes VGGT with the elimination of redundant intermediate features. Finally, part (c) illustrates that, with an appropriate joint pose and 3DGS optimization strategy, a photorealistic rendering can be realized.

## Abstract

We study the problem of applying 3D Foundation Models (3DFMs) to dense Novel View Synthesis (NVS). Despite significant progress in Novel View Synthesis powered by NeRF and 3DGS, current approaches remain reliant on accurate 3D attributes (e.g., camera poses and point clouds) acquired from Structure-from-Motion (SfM), which is often slow and fragile in low-texture or low-overlap captures. Recent 3DFMs showcase orders of magnitude speedup over the traditional pipeline and great potential for online NVS. But most of the validation and conclusions are confined to sparse-view settings. Our study reveal that naively scaling 3DFMs to dense views encounters two fundamental barriers: dramatically increasing VRAM burden and imperfect outputs that degrade initialization-sensitive 3D training. To address these barriers, we introduce **VGGT-X**, incorporating a memory-efficient VGGT implementation that scales to 1,000+ images, an adaptive global alignment for VGGT output enhancement, and robust 3DGS training practices. Extensive experiments show that these measures substantially close the fidelity gap with COLMAP-initialized pipelines, achieving state-of-the-art results in dense COLMAP-free NVS and pose estimation. Additionally, we analyze the causes of remaining gaps with COLMAP-initialized rendering, providing insights for the future development of 3D foundation models and dense NVS.

## 1 Introduction

Novel View Synthesis (NVS) reconstructs a 3D scene from multi-view images to render photorealistic novel views. Implicit representations like Neural Radiance Fields (NeRF) (Mildenhall et al., 2021) set a new standard in rendering fidelity, while recent explicit 3D Gaussian Splatting (3DGS)

(Kerbl et al., 2023) revolutionizes the area by enabling realistic rendering with real-time speed. Both families, however, typically require tens of minutes to train and depend on accurate initialization from external sensors or reconstruction pipelines (e.g., COLMAP (Schönberger & Frahm, 2016)), which incurs expensive hardware or additional minutes-to-hours overhead (Li et al., 2025).

Recent 3D Foundation Models (3DFMs) offer a promising alternative by dramatically accelerating components of the pipeline. For example, VGGT can infer camera poses and depth for 200 images in 10s (Wang et al., 2025a), and Anysplat produces 3DGS from 64 views in 5s (Jiang et al., 2025a), suggesting orders-of-magnitude speedups over classic pipelines. Yet these methods are largely demonstrated in sparse-view regimes (tens of images), leaving open the question: **what would happen if 3DFMs are applied to dense NVS?**

This work investigates applying 3DFMs to dense NVS and identifies two central obstacles. First, 3DFM computation and memory cost increase dramatically with the number of views (e.g., VRAM of VGGT rises from 5.6 GB to 40.6 GB when input rises from 20 to 200 views (Wang et al., 2025a)), making direct dense inference of 3DGS properties infeasible on commercial GPUs. Second, even when used as a drop-in replacement for traditional reconstruction pipelines like COLMAP, 3DFM outputs exhibit higher noise levels. Such noise undermines the learning of initialization-sensitive 3D primitives and leads to significant degradation in rendering quality.

To remove the obstacles and explore the answer to the question, we take VGGT (Wang et al., 2025a) as a representative 3DFM and pursue two directions. On the 3DFM side, we remove redundant feature caching, reduce numeric precision, and adopt batched frame-wise operations to losslessly scale VGGT inference to 1,000+ images (see part (b) of Fig. 1). On the 3DGS side, we study the effect of initializing 3DGS with VGGT outputs. Tab. 4 shows substantial degradation under naïve initialization. We further investigate whether the mitigation strategy exists. We propose an efficient adaptive global alignment under epipolar constraints to refine VGGT predictions. Besides, we adopt MCMC-3DGS (Kheradmand et al., 2024) and joint pose optimization to increase robustness to noisy initialization, along with a point-cloud initialization strategy through comparative analysis. Through these approaches, we largely mitigate the fidelity gap and obtain state-of-the-art rendering under COLMAP-free settings. We also analyze remaining discrepancies with COLMAP-initialized training, including overfitting and generalization problems, and offer concrete directions for stronger 3DFMs and more robust NVS training.

In summary, our contributions are fourfold:

- We identify and analyze the key problems that prevent current 3DFMs from scaling to dense NVS.
- We explore and reveal how the key problems can be alleviated by introducing VGGT-X, a memory-efficient VGGT implementation combined with an adaptive global alignment and 3DGS training practices tailored to imperfect initialization.
- We analyze the residual gap to COLMAP-initialized pipelines and provide insights to strengthen future 3DFMs and NVS training.
- Extensive experiments confirm our state-of-the-art performance in both pose estimation and COLMAP-free NVS.

## 2 RELATED WORKS

### 2.1 NOVEL VIEW SYNTHESIS

**Novel view synthesis (NVS)** seeks to generate photorealistic images from novel viewpoints given a set of input images captured from different perspectives of a 3D scene. This task fundamentally relies on reconstructing a faithful 3D representation of the scene. A landmark in this field is **Neural Radiance Fields (NeRF)** (Mildenhall et al., 2021), which employs multi-layer perceptrons (MLPs) to implicitly encode scene geometry and appearance. Subsequent works have advanced NeRF along multiple directions, including improved reflectance modeling (Verbin et al., 2022; Attal et al., 2023), anti-aliasing techniques (Barron et al., 2021; 2022), and acceleration of both training and inference (Zhang et al., 2023; Müller et al., 2022; Yu et al., 2021). More recently, **3D Gaussian Splatting (3DGS)** (Kerbl et al., 2023) has emerged as a powerful alternative, offering substantial efficiency

gains while preserving high rendering quality. Building on this foundation, recent research has extended 3DGS to large-scale scene reconstruction (Lin et al., 2024; Liu et al., 2024; 2025), compact storage and transmission (Fan et al., 2024a; Lee et al., 2024), and artifact mitigation (Yu et al., 2024; Ye et al., 2024; Radl et al., 2024). Despite these advances, NVS methods still require accurate camera parameters, and 3DGS in particular remains highly sensitive to the quality of the initial point cloud. Inaccurate poses or noisy geometry often result in visual artifacts and geometric misalignments in the synthesized views.

## 2.2 3D FOUNDATION MODELS

**3D foundation models** aim to infer fundamental 3D attributes—such as camera parameters, point clouds, depth maps, point tracks, or even neural radiance fields—directly from image collections. Current approaches are broadly instantiated through two architectural paradigms: diffusion-based models (Ho et al., 2020) and feed-forward ViT-based models (Dosovitskiy et al., 2021). Based on input types, the 3D foundation models can be categorized into **4 types** (Cong et al., 2025). **(i)** For **uncalibrated image pairs**, DUSt3R (Wang et al., 2024b) and its successors (Leroy et al., 2024; Fan et al., 2024b; Zhang et al., 2025a; Ye et al., 2025; Lu et al., 2025; Smart et al., 2024; Chen et al., 2025) predict point clouds (with auxiliary properties such as confidence) within the coordinate frame of the first camera. Through additional correspondence matching and reprojection loss optimization, these local geometries can be aligned into a consistent global frame (Duisterhof et al., 2025). **(ii)** For **unordered multi-view image collections**, models such as (Yang et al., 2025; Wang et al., 2025a;c; Fang et al., 2025) employ inter- and intra-view cross-attention to directly produce globally consistent poses and geometry. **(iii)** For **image streams**, models like Spann3R (Wang & Agapito, 2025) and CUT3R (Wang et al., 2025b) predict next-frame geometry by leveraging current features and temporal memory, while diffusion-based approaches (Team et al., 2025; Jiang et al., 2025b; Xu et al., 2025) cast geometry estimation as a conditional generative process. **(iv)** For **uncalibrated sparse views**, FLARE (Zhang et al., 2025b) adopts a cascaded, feed-forward pipeline that first regresses camera poses and then conditions global geometry and appearance estimation. Despite rapid progress, most existing models incur substantial computational overhead and exhibit degraded performance when scaled to hundreds or thousands of images. Our study aims to address this gap and provide new insights into the development of scalable 3D foundation models.

## 2.3 3D RADIANCE FIELD LEARNING WITH POSE OPTIMIZATION

To mitigate the dependence on accurate camera poses, recent NVS approaches have explored a variety of strategies. A widely adopted solution is the joint optimization of camera parameters alongside the neural radiance field, often complemented by multi-view correspondence losses (Wang et al., 2021; Jeong et al., 2021). Methods such as NoPe-NeRF (Bian et al., 2023) and SPARF (Truong et al., 2023) incorporate depth supervision, whereas (Bian et al., 2024; Huang et al., 2025b) employ MLPs to regress pose updates, enhancing robustness and exploiting global scene context. NeRF-based techniques further investigate strategies to mitigate sub-optimal convergence caused by high-frequency positional embeddings (Lin et al., 2021; Chng et al., 2022; Xia et al., 2022). In the context of 3DGS, MCMC-3DGS (Kheradmand et al., 2024) enhances robustness to initialization by reformulating the Gaussian Splatting update mechanism, while (Fu et al., 2024; Chen et al., 2024; Ji & Yao, 2025) perform incremental local geometry reconstruction and pose refinement for unposed image sequences. More recently, approaches leveraging 3D foundation models or tracking models (Huang et al., 2025a;b; Shi et al., 2025) have been proposed to efficiently obtain high-quality initializations of poses and geometry. Despite these advances, a notable performance gap remains compared to COLMAP-initialized optimization, and scaling these methods to large image collections remains largely unexplored. Our work aims to advance this frontier, providing insights into training photorealistic neural radiance fields from imperfectly registered poses and point clouds.

## 3 METHOD

### 3.1 PRELIMINARY

**3D Gaussian Splatting** (Kerbl et al., 2023) models a 3D scene using a collection of ellipsoids parameterized by 3D Gaussian distributions, i.e., $\mathbb{G} = \{\mathcal{G}_i \mid i = 1, \ldots, N_G\}$. Each Gaussian is as-

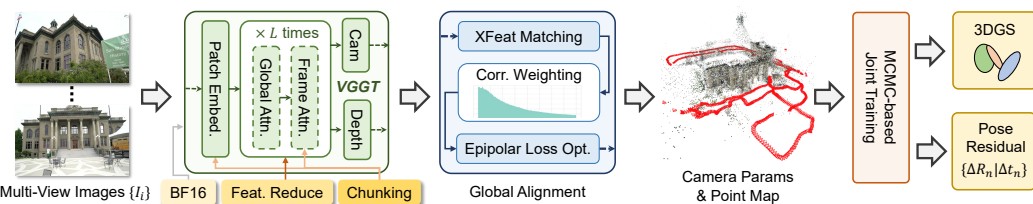

Figure 2: Overall pipeline of our model.

sociated with learnable attributes, including its center $\boldsymbol{\mu_i} \in \mathbb{R}^{3\times1}$, covariance matrix $\boldsymbol{\Sigma_i} \in \mathbb{R}^{3\times3}$, opacity $\sigma_i \in [0,1]$, and spherical harmonics (SH) features $\boldsymbol{f_i} \in \mathbb{R}^{3\times16}$ for view-dependent appearance modeling. The covariance matrix is further decomposed into a scaling matrix $\mathbf{S_i}$ and a rotation matrix $\mathbf{R_i}$, such that $\boldsymbol{\Sigma_i} = \mathbf{R_i}\mathbf{S_i}\mathbf{S_i}^T\mathbf{R_i}^T$. For a given pixel $p$, the color $\boldsymbol{c_p}$ is obtained via alpha blending. Given a ground-truth image $\mathbf{I}$, the optimization of 3DGS is driven by the total loss $\mathcal{L}_{\text{total}}$, defined as the weighted combination of the L1 loss $\mathcal{L}_1$ and the D-SSIM loss $\mathcal{L}_{\text{SSIM}}$. To mitigate under- or over-reconstruction, 3DGS employs a heuristic adaptive density control strategy guided by the view-space position gradient $\nabla_{\text{densify}} = \partial\mathcal{L}/\partial\boldsymbol{\mu_i}$. Gaussians with gradients exceeding a predefined threshold are either cloned or split. We refer readers to the original paper (Kerbl et al., 2023) for additional details.

**3DGS-MCMC** (Kheradmand et al., 2024) improves 3DGS in both rendering fidelity and robustness to noisy initialization. The key insight is that the optimization of 3DGS can be reformulated as a Stochastic Gradient Langevin Dynamics (SGLD) update:

$$\mathcal{G} \leftarrow \mathcal{G} - \lambda_{\text{lr}} \cdot \nabla_{\mathcal{G}}\mathbb{E}_{\mathbf{I}\sim\mathcal{I}}[\mathcal{L}_{\text{total}}(\mathcal{G}; \mathbf{I})] + \lambda_{\text{noise}} \cdot \boldsymbol{\epsilon}, \tag{1}$$

where $\lambda_{\text{lr}}$ and $\lambda_{\text{noise}}$ denote hyperparameters that control the learning rate and the magnitude of stochastic exploration, respectively, and $\boldsymbol{\epsilon}$ represents noise sampled for exploration. To mitigate the dependency on precise initialization, we adopt 3DGS-MCMC as our baseline for NVS.

### 3.2 MEMORY-EFFICIENT VGGT IMPLEMENTATION

As illustrated in Fig. 2, the network structure of VGGT comprises three main components: per-frame DINO-based patch embedding extractor, stacked transformer layers alternating between global and frame-wise attention (i.e., AA layers), and a decoder for camera parameter regression and dense predictions (Wang et al., 2025a). Although VGGT contains 24 AA layers, only the output features from layers 4, 11, 17, and 23 are utilized for dense prediction. To eliminate redundancy, we discard intermediate outputs from other layers, thereby reducing VRAM consumption. This modification increases image throughput from 150 to 600 images, and we refer to this variant as VGGT−.

Another source of redundancy lies in data precision. While automatic mixed precision is enabled, the majority of operations and tensor storage still default to Float32. We observe that switching to BFloat16, except for MLP in heads, introduces no noticeable degradation in performance. In contrast, it reduces the peak GPU memory usage by up to 74%, leading to a substantial improvement in inference throughput. Moreover, since both DINO feature extraction and frame-wise attention involve only intra-frame computation, frames can be processed asynchronously. Consequently, $N$ input images can be divided into $\lceil N/S \rceil$ chunks, which are sequentially processed. By selecting an appropriate chunk size $S$, peak memory usage in these modules can be effectively controlled. For convenience, this version is named as VGGT−−.

### 3.3 CAMERA PARAMETERS GLOBAL ALIGNMENT (GA)

After the feedforward inference of VGGT, we obtain estimated camera parameters $\{\mathcal{K}_n, \mathcal{R}_n, t_n\}_N$, where for the $n$-th camera, $\mathcal{K}_n$ denotes the intrinsic matrix, while $\mathcal{R}_n$ and $t_n$ represent the rotation matrix and translation vector of the extrinsic matrix, respectively. These parameters can be refined using image correspondences by minimizing the epipolar distance loss:

$$\mathcal{L}_{EG} = \sum_m \sum_k w_k\, e_{m,k} \Big/ \sum_m \sum_k w_k, \quad e_{m,k} = x'_k F_m x_k, \tag{2}$$

where $e_{m,k}$ is the epipolar distance for the $k$-th correspondence in the $m$-th image pair, $x'_k$ and $x_k$ are the corresponding keypoints, and $F_m$ is the fundamental matrix derived from the paired cameras. The weights $w_k$ reflect the reliability of each correspondence, making their estimation crucial for effective optimization.

Not all $C_N^2$ image pairs have overlapping fields of view. Following (Jeong et al., 2021), we restrict candidate pairs to those with view angles below a certain threshold. For these pairs, VGGT's tracking head can provide correspondences and confidence scores. However, as shown in Tab. 3, these predictions are insufficiently reliable for camera refinement. We therefore adopt XFeat (Potje et al., 2024), a recent neural feature matcher known for its efficiency. While XFeat provides accurate matches, it does not supply correspondence weights $w_k$. Using VGGT's depth confidence as a proxy also proves suboptimal (cf. Tab. 3).

To address this issue, we propose an adaptive weighting strategy. Intuitively, when both the 3D foundation model and the matching model provide reliable estimates, most $e_{m,k}$ values should cluster near zero, and such correspondences should be assigned higher weights. Conversely, correspondences with large $e_{m,k}$ are more likely to be outliers and should be down-weighted. The "Global Alignment" panel in Fig. 2 illustrates a typical histogram of $e_{m,k}$ with the x-axis limited to $[0, 20]$. As observed, $e_{m,k}$ exhibits a long-tail distribution, which aligns naturally with this intuition. Accordingly, we first compute $e_{m,k}$ using VGGT-predicted camera parameters as defined in Eq. (2), and then estimate the adaptive weights as:

$$w_k = \left( \frac{f(e_{m,k})}{\text{Avg}(f(e_{m,k}))} \right)^\alpha,$$
(3)

where $f$ is the probability density function approximated via histogram, $\text{Avg}(f(e_{m,k}))$ denotes the average density over all $e_{m,k}$, and $\alpha$ is empirically set to $0.5$. As validated in Tab. 3, this weighting scheme enables more efficient convergence during camera optimization.

Finally, we adapt the learning rate to different convergence regimes. When VGGT's initialization is accurate, a small learning rate suffices for fine alignment. However, in challenging cases, such a setting fails to provide adequate updates. To adaptively control learning, we use the median epipolar distance as an indicator and adjust the learning rate according to the following empirical rule:

$$\text{lr} = \begin{cases} \text{lr}_0, & \text{if Median}(e_{m,k}) < b_1, \\ \text{lr}_1, & \text{if } b_1 < \text{Median}(e_{m,k}) < b_2, \\ \text{lr}_2, & \text{if Median}(e_{m,k}) > b_2, \end{cases}$$
(4)

where $\text{lr}_0$, $\text{lr}_1$, $\text{lr}_2$ and the bounds $b_1$, $b_2$ are specified in Sec. 4.1. As shown in Tab. 3, this adaptive strategy is critical for ensuring robust convergence in camera parameter optimization.

### 3.4 3DGS Training with Imperfect Poses

The global alignment procedure in Sec. 3.3 substantially improves the accuracy of estimated camera parameters, thereby facilitating convergence of 3DGS training. Nonetheless, the performance gap relative to COLMAP remains, which is detrimental for initialization-sensitive models such as vanilla 3DGS (cf. Tab. 4). To mitigate this issue, we adopt MCMC-3DGS, which offers improved robustness under noisy or imperfect poses.

In addition, we adopt a joint optimization scheme in which residual camera poses are optimized alongside Gaussian parameters under photometric supervision. Concretely, we estimate the residual translation $\Delta t_n \in \mathbb{R}^3$ and residual rotation $\Delta r_n \in \mathbb{R}^6$. Following (Zhou et al., 2019), the 6D rotation representation $\Delta r_n$ is converted into a residual rotation matrix $\Delta \mathcal{R}_n \in \mathbb{R}^{3 \times 3}$, which is then applied to refine $\mathcal{R}_n$. In addition, we leverage the correspondence weights introduced in Sec. 3.3 to select reliable initialization points, providing a stronger starting configuration for 3DGS training. As shown in Tab. 4, this strategy leads to consistently improved performance.

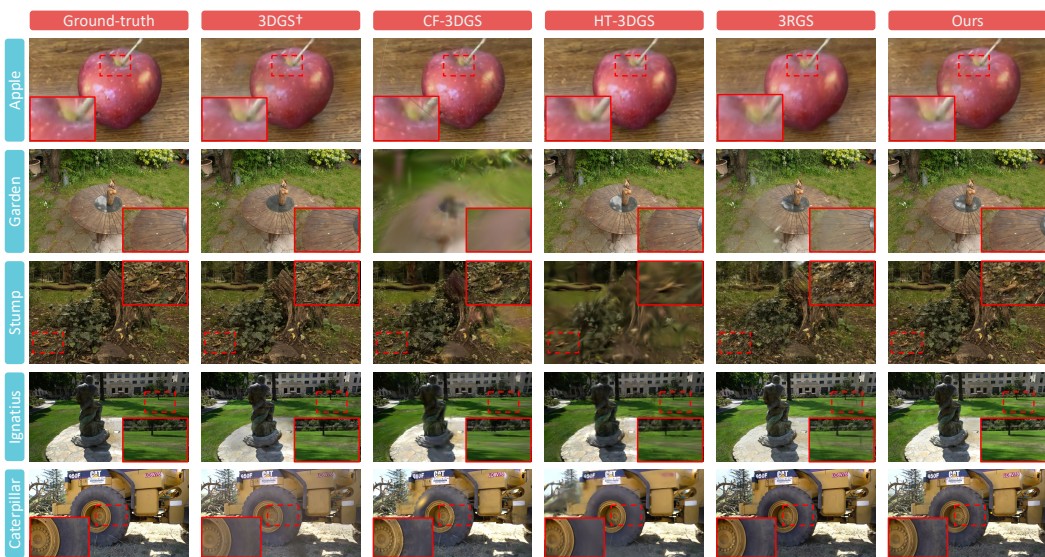

Figure 3: Qualitative comparison of rendering results. 3DGS$^\dagger$ here means 3DGS trained with COLMAP initialization, and is mainly for reference. Here, *Apple* is from *CO3Dv2* dataset, *Garden* and *Stump* are from *MipNeRF360* dataset, *Ignatius* and *Caterpillar* are from *TnT* dataset.

## 4 EXPERIMENTS

### 4.1 EXPERIMENTAL SETUP

**Datasets & Metrics.** We evaluate our model on widely used multi-view reconstruction benchmarks, including *MipNeRF360* (Barron et al., 2022), *Tanks and Temple (TnT)* (Knapitsch et al., 2017), and *CO3Dv2* (Reizenstein et al., 2021), with maximum image sequence lengths of 311, 1106, 202 and scene numbers of 9, 5, 5, respectively. *MipNeRF360* is employed for our ablation studies. We follow (Wang et al., 2025a) for pose and point map estimation. **Pose accuracy** is evaluated using the standard AUC@30 metric, which integrates Relative Translation Error (RTE) and Relative Rotation Error (RRE). RTE and RRE compute the relative angular errors in translation and rotation for each image pair. We note that *AUC@30 is not order-invariant* and introduce a minor modification to address this; further details are provided in the Appendix C. **Point map quality** is measured using Chamfer Distance, alongside accuracy and completeness metrics. For multi-view reconstruction, we adhere to the dataset splits and training view resolutions reported in prior works (Kerbl et al., 2023; Fu et al., 2024). **Rendering quality** is assessed via PSNR, SSIM, and LPIPS. For computational efficiency, we report both runtime and VRAM usage measured on a 40G A100 GPU.

**Implementation Details.** In our experiments, the chunk size $S$ for the frame-wise operation described in Sec. 3.2 is set to 128. For the global alignment procedure in Sec. 3.3, the angle-of-view threshold is set to 30 degrees. The learning rates $\text{lr}_0$, $\text{lr}_1$, and $\text{lr}_2$ are configured to $5 \times 10^{-4}$, $1 \times 10^{-3}$, and $1 \times 10^{-2}$, respectively, while the parameters $b_1$ and $b_2$ are set to 2.5 and 7.5. The maximum number of correspondences per image pair is limited to 4096, and the optimization is run for 300 iterations. When extracting COLMAP results, only matched points with weights exceeding 0.3 are retained. During MCMC-3DGS training, the maximum number of Gaussians per scene is matched to that of the vanilla 3DGS to ensure fairness. Pose embeddings are initialized with a learning rate of $1 \times 10^{-4}$ and decayed exponentially by a factor of 0.1, while the learning schedule for other 3DGS attributes follows (Kheradmand et al., 2024). During rendering quality assessment, we would freeze trained Gaussians and tune the pose embedding for the test view and minimize the photometric loss, following practice of (Huang et al., 2025b). The tuning iteration is 10,000 for *TnT* and 5,000 for other datasets, while the other setting of learning schedule aligns with the training progress.

**Baselines.** For 3D key attributes prediction, we compare the performance of MASt3R-SfM (Duisterhof et al., 2025), $\pi^3$ (Wang et al., 2025c), and VGGT (Wang et al., 2025a). For MASt3R-SfM, we employ the retrieval mode in scene graph construction to achieve a balance between accuracy and

Table 1: Comparison with SOTA methods on rendering quality. † means initialized with COLMAP. Note that for fairness, 3RGS is also trained on predictions from our VGGT−− with GA. The best performance of each part is in **bold**.

| Model | MipNeRF360 | | | Tanks and Temple | | | CO3Dv2 | | |
|---|---|---|---|---|---|---|---|---|---|
| | SSIM↑ | PSNR↑ | LPIPS↓ | SSIM↑ | PSNR↑ | LPIPS↓ | SSIM↑ | PSNR↑ | LPIPS↓ |
| 3DGS† | 0.8148 | 27.39 | 0.1849 | 0.8509 | 24.85 | 0.1550 | 0.9379 | 32.58 | 0.0954 |
| MCMC† | 0.8357 | 27.91 | 0.1536 | 0.8674 | 25.76 | 0.1391 | 0.9407 | 33.21 | 0.0968 |
| MCMC | 0.5484 | 22.19 | 0.2822 | 0.6789 | 21.42 | 0.2778 | 0.7121 | 25.71 | 0.2008 |
| CF-3DGS | 0.2344 | 12.38 | 0.7186 | 0.3914 | 12.19 | 0.6082 | 0.6110 | 20.18 | 0.4354 |
| HT-3DGS | 0.3796 | 14.79 | 0.6691 | 0.4508 | 13.83 | 0.5850 | 0.8326 | 28.28 | 0.2298 |
| 3RGS | 0.7128 | 25.39 | 0.2158 | 0.7497 | 21.47 | 0.3002 | 0.8781 | 31.07 | 0.1283 |
| Ours | **0.7821** | **26.40** | **0.1774** | **0.8419** | **24.77** | **0.1676** | **0.9105** | **31.85** | **0.1128** |

Table 2: Comparison with SOTA methods on pose estimation. The units for RRE and RTE are degrees. Note that for fairness, 3RGS is also trained on predictions from our VGGT−− with GA. The best performance of each part is in **bold**.

| Model | MipNeRF360 | | | Tanks and Temple | | | CO3Dv2 | | |
|---|---|---|---|---|---|---|---|---|---|
| | RRE↓ | RTE↓ | AUC@30↑ | RRE↓ | RTE↓ | AUC@30↑ | RRE↓ | RTE↓ | AUC@30↑ |
| MASt3R-Sfm | 17.18 | 10.25 | 0.718 | 21.02 | 14.10 | 0.687 | 11.72 | 15.32 | 0.618 |
| $\pi^3$ | 3.244 | 3.470 | 0.889 | OOM | OOM | OOM | **0.924** | **1.719** | **0.956** |
| VGGT−− | 1.094 | 1.759 | 0.951 | 2.034 | 1.891 | 0.953 | 3.035 | 4.659 | 0.841 |
| VGGT−−, +GA | 0.678 | 0.686 | 0.986 | 1.783 | 1.479 | 0.967 | 2.002 | 2.811 | 0.906 |
| CF-3DGS | 104.0 | 56.45 | 0.001 | 110.9 | 55.20 | 0.006 | 15.2 | 21.5 | 0.336 |
| HT-3DGS | 93.69 | 56.55 | 0.003 | 100.0 | 51.87 | 0.010 | 12.30 | 12.25 | 0.501 |
| 3RGS | 0.605 | 0.484 | 0.991 | 4.855 | 6.762 | 0.846 | 1.972 | 2.583 | 0.911 |
| Ours | **0.601** | **0.484** | **0.992** | **1.738** | **1.259** | **0.971** | 1.984 | 2.687 | 0.909 |

efficiency. For COLMAP-free 3DGS training, we consider CF-3DGS (Fu et al., 2024), HT-3DGS (Ji & Yao, 2025), 3RGS (Huang et al., 2025b), and MCMC-3DGS (Kheradmand et al., 2024). To ensure a fair comparison, we replace the initial poses and point cloud in 3RGS with our globally aligned, higher-accuracy results.

## 4.2 Comparison with SOTA Methods

In Tab. 1, we compare rendering performance against recent advances and include results with COLMAP initialization as an upper-bound reference. Our model achieves state-of-the-art performance, as further confirmed by the qualitative results in Fig. 3, which show that our method more effectively suppresses blurry artifacts and floaters while preserving fine-grained textures. It is worth noting that the rendering quality of CF-3DGS on *CO3Dv2* is noticeably worse than reported in its original paper, likely due to reproducibility issues documented in its repository[1].

In Tab. 2, we compare pose estimation accuracy. The results demonstrate that both our global alignment and joint optimization strategies consistently improve performance, surpassing all previous approaches that jointly optimize poses and 3DGS. We also evaluate the pose accuracy of 3D foundation models and provide trajectory visualizations in Fig. 4. Our model exhibits closer alignment with ground-truth trajectories, achieving the highest accuracy on *MipNeRF360* and *TnT*, and ranking second on *CO3Dv2*.

## 4.3 Ablation

First, we ablate the effect of different modules on 3D key attribute estimation. The primary reduction in computational overhead comes from redundant feature elimination and precision adjustment,

---

[1] https://github.com/NVlabs/CF-3DGS/issues/7

Table 3: Ablation on model components in pose and point map estimation. The experiments are conducted on *MipNeRF360* (Barron et al., 2022). "-XFeat" here means replacing XFeat with tracking predicted by VGGT itself. "- PDF Weight" means using confidence predicted by VGGT to replace adaptive weight proposed in Sec. 3.3. Computation costs are evaluated on 40G A100.

| Model | Pose Estimation | | | Point Map Estimation | | | Cost | |
| --- | --- | --- | --- | --- | --- | --- | --- | --- |
| | RRE(°)↓ | RTE(°)↓ | AUC@30↑ | Acc.↓ | Comp.↓ | Overall↓ | T(min) | Mem.(GB) |
| VGGT | OOM | OOM | OOM | OOM | OOM | OOM | OOM | OOM |
| VGGT− | 1.090 | 1.740 | 0.951 | 0.064 | 0.051 | 0.058 | **0.98** | 28.87 |
| VGGT−− | 1.094 | 1.759 | 0.951 | **0.063** | 0.050 | 0.057 | 1.29 | **9.66** |
| VGGT−−, +BA | **0.640** | **0.392** | **0.994** | 0.064 | **0.037** | **0.050** | 157 | 24.26 |
| VGGT−−, +GA | **0.652** | **0.643** | **0.988** | 0.069 | 0.039 | 0.054 | **1.78** | **11.12** |
| - XFeat | 2.096 | 2.250 | 0.920 | 0.220 | 0.184 | 0.202 | 4.46 | 13.49 |
| - Adaptive LR | 0.732 | 0.751 | 0.984 | **0.064** | **0.037** | **0.051** | 1.78 | 11.12 |
| - PDF Weight | 2.705 | 2.970 | 0.892 | 0.108 | 0.058 | 0.083 | 1.88 | 11.12 |
| - Rand Order | 0.681 | 0.691 | 0.986 | 0.068 | 0.040 | 0.054 | 1.78 | 11.12 |

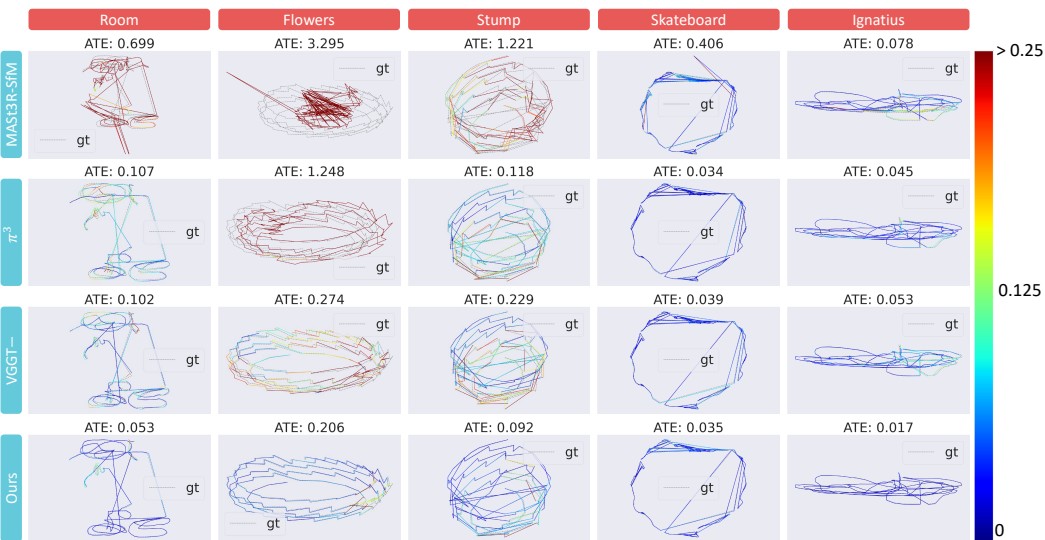

Figure 4: Qualitative comparison of estimated trajectories. *Skateboard* comes from *CO3Dv2* dataset, *Ignatius* comes from *TnT* dataset, *Room*, *Flowers*, and *Stump* come from *MipNeRF360* dataset. Here we also report the Root Mean Square Error (RMSE) of the Absolute Trajectory Error (ATE) (in meters) (Matsuki et al., 2024). The color bar indicates trajectory distance. We recommend zooming in for better details.

which together lower VRAM usage by 83% on *MipNeRF360*. Batched attention further reduces memory by over 1 GB when scaling to more than 800 images. Combined these modifications together, the inference throughput is pushed to 1000+ images, as shown in Fig. 1. Noticeably, these optimizations have only a negligible impact on prediction accuracy, as indicated in Tab. 3.

Second, we examine strategies to enhance VGGT output quality. Replacing XFeat with the VGGT tracking head decreases AUC@30 by 6.8 points and increases Chamfer Distance by nearly fourfold. Similarly, leveraging VGGT-derived depth confidence to reweight XFeat correspondences results in substantial performance degradation. In contrast, incorporating our adaptive learning rate yields consistently higher accuracy. Moreover, we observe that with permutation-equivariant AUC@30, a random input order still yields a slight performance gain, consistent with the findings of (Wang et al., 2025c). Besides, we also scale the official Bundle Adjustment (BA) strategy to hundreds of images by applying our architectural optimizations to VGG-SfM (Wang et al., 2024a). While this achieves higher accuracy, it requires over two hours to complete, and as shown in Tab. 4, its initialization does not improve NVS quality, confirming the superior efficiency of our strategy.

Table 4: Ablation on model components in multi-view reconstruction. The experiments are conducted on *MipNeRF360* (Barron et al., 2022). The best performance of each metric is in **bold**.

| Model | Initialization | | Train Set | | | Test Set | | |
|---|---|---|---|---|---|---|---|---|
| | Pose | Point Map | SSIM↑ | PSNR↑ | LPIPS↓ | SSIM↑ | PSNR↑ | LPIPS↓ |
| 3DGS | COLMAP | COLMAP | 0.8869 | 29.58 | 0.1427 | 0.7194 | 27.39 | 0.1849 |
| MCMC | COLMAP | COLMAP | 0.9041 | 30.17 | 0.1231 | 0.8357 | 27.91 | 0.1536 |
| +Pose Opt. | COLMAP | COLMAP | 0.9042 | 30.19 | 0.1228 | 0.8359 | 27.95 | 0.1537 |
| 3DGS | VGGT−− | Rand. 500K | 0.7284 | 23.95 | 0.2830 | 0.5321 | 21.10 | 0.3466 |
| 3DGS | VGGT−−, +GA | Rand. 500K | 0.7538 | 25.00 | 0.2471 | 0.5675 | 22.23 | 0.3058 |
| MCMC | VGGT−−, +GA | Rand. 500K | 0.8178 | 26.41 | 0.1974 | 0.5563 | 22.37 | 0.2795 |
| +Pose Opt. | VGGT−−, +GA | Rand. 500K | 0.8965 | 29.25 | **0.1229** | 0.7731 | 26.28 | 0.1823 |
| +Pose Opt. | VGGT−−, +GA | Rand. 500K | 0.8965 | 29.25 | **0.1229** | 0.7731 | 26.28 | 0.1823 |
| +Pose Opt. | VGGT−−, +GA | Filtered. 500K | 0.8794 | 28.85 | 0.1473 | 0.7620 | 25.88 | 0.2005 |
| +Pose Opt. | VGGT−−, +GA | Matched Points | **0.8966** | **29.59** | 0.1314 | **0.7821** | **26.40** | **0.1774** |
| +Pose Opt. | VGGT−, +BA | VGGT−, +BA | 0.8948 | 29.23 | 0.1301 | 0.7765 | 26.33 | 0.1786 |

Finally, we ablate design choices for training high-quality 3DGS. As shown in Tab. 4, MCMC is more effective than vanilla 3DGS under imperfect initialization, and pose optimization proves essential for stable convergence and high rendering quality. Among initialization strategies, point clouds derived from high-confidence correspondences achieve the best performance. Limited by pages, we put additional ablations in Tab. 5 in the Appendix.

## 4.4 DISCUSSION

Although our model achieves state-of-the-art performance, a noticeable gap remains compared to 3DGS trained with COLMAP initialization, as shown in Tab. 1. Interestingly, Tab. 4 reveals that on the training set, our model even surpasses COLMAP-initialized 3DGS in rendering quality, yet its performance on the test set lags behind, suggesting a clear overfitting issue. This highlights the inherently ill-posed nature of the problem. And without reliable initialization, the optimization process is prone to getting trapped in local minima of the highly non-convex loss landscape. We also experimented with adding depth supervision (cf. Tab. 5), but found little improvement.

Besides, as illustrated in Tab. 2, even after joint optimization, pose accuracy still falls short of COLMAP. We further compare the learned camera pose residuals with ground-truth. Limited by page, the visualization is included in Fig. 5 of Appendix A. We observe that while most residuals cluster near zero—indicating accurately predicted poses—the model struggles to sufficiently correct poses with large deviations. Another noteworthy finding in Tab. 2 is that although VGGT substantially outperforms $\pi^3$ on *MipNeRF360*, it is surpassed on *CO3Dv2* by a considerable margin. This discrepancy suggests that the generalization ability of 3D foundation models remains an open challenge.

## 5 CONCLUSIONS

In this paper, we investigated the potential of applying 3D Foundation Models to dense novel view synthesis and identified two key challenges: the poor scalability in computational overhead and insufficient prediction accurary for subsequent radiance field fitting. To address these obstacles, we introduced VGGT-X, which integrates a memory-efficient VGGT implementation, adaptive global alignment, and robust 3DGS training strategies. Our approach substantially narrows the performance gap with COLMAP-initialized counterparts. Beyond these improvements, our analysis also sheds light on the remaining limitations and outlines promising directions for advancing both 3DFMs and NVS frameworks. We hope our findings provide valuable insights toward building fast, reliable, and fully COLMAP-free dense NVS systems.

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

## A  BAD CASE ANALYSIS

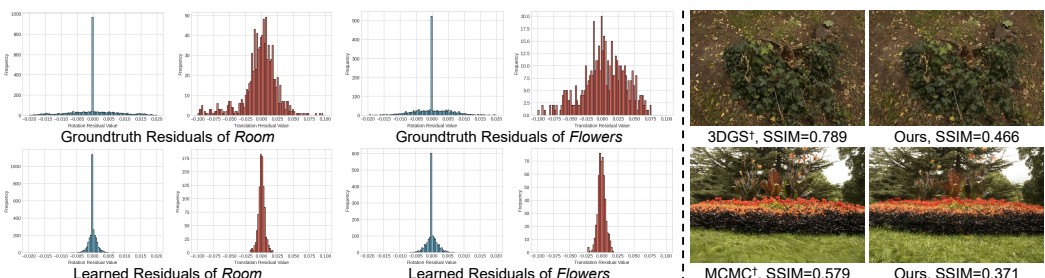

Figure 5: Bad case analysis. The blue and red histograms respectively correspond to rotation and translation residual distribution. The right part shows blurry artifacts caused by inaccurate poses.

This section provides a visualization of bad cases. Detailed analysis is included in Sec. 4.4.

## B  ADDITIONAL ABLATIONS

Table 5: Additional ablation on model components in multi-view reconstruction. The experiments are conducted on *MipNeRF360* (Barron et al., 2022). This table showcases the aborted model designs. The best performance of each metric is in **bold**. The "Baseline" denotes MCMC-3DGS equipped with pose embedding. The modification of each following row is independent of the others.

| Model | Initialization | | Train Set | | | Test Set | | |
|---|---|---|---|---|---|---|---|---|
| | Pose | Point Map | SSIM↑ | PSNR↑ | LPIPS↓ | SSIM↑ | PSNR↑ | LPIPS↓ |
| Baseline | Ours | Matched Points | 0.8966 | 29.59 | 0.1314 | **0.7821** | **26.40** | 0.1774 |
| w MLP | Ours | Matched Points | 0.8753 | 28.60 | 0.1436 | 0.7492 | 25.83 | 0.1934 |
| w depth | Ours | Matched Points | 0.8851 | 28.85 | 0.1415 | 0.7628 | 25.94 | 0.1954 |
| w $2\times$ pose lr | Ours | Matched Points | **0.9044** | **30.02** | **0.1243** | 0.7740 | 26.16 | **0.1737** |
| w Epi. Loss | Ours | Matched Points | 0.8964 | 29.55 | 0.1318 | 0.7795 | 26.31 | 0.1780 |
| w Epi. Loss | VGGT* | VGGT* | 0.8499 | 27.21 | 0.1806 | 0.6440 | 23.22 | 0.2572 |

Here we provide additional ablation studies in Tab. 5. We experimented with design choices like MLP-based pose embedding learning (Huang et al., 2025b), epipolar loss during 3DGS training, and depth supervision. But none of them bring clear benefits to the rendering quality. We also tried to double learning rate and encourage to learn a broader distribution, but it turns out to aggregate the overfitting phenomenon. Moreover, the last row of Tab. 5 shows that integrating global alignment into GS training, rather than treating it as a separate process, leads to suboptimal results. Therefore, we adopt global alignment as an independent component.

## C  PERMUTATION-EQUIVARIANT AUC@30

In this section, we analyze why the conventional AUC@30 metric is sensitive to the input image order and propose a simple yet effective modification to address this issue. AUC@30 first computes relative poses for all $C_N^2$ image pairs. Comparing the relative poses from ground truth and predictions, the relative rotation and translation errors can be derived for AUC@30 calculation. Specifically, for two images indexed by $i$ and $j$ (with $i < j$) and their corresponding extrinsics, the relative pose is computed as:

$$\Delta E_{ij} = E_i^{-1} E_j = \begin{pmatrix} R_i^T & -R_i^T t_i \\ 0 & 1 \end{pmatrix} \begin{pmatrix} R_j & t_j \\ 0 & 1 \end{pmatrix} = \begin{pmatrix} R_i^T R_j & R_i^T (t_j - t_i) \\ 0 & 1 \end{pmatrix}. \tag{5}$$

However, if the image order is permuted and $j$ precedes $i$, the relative pose becomes:

$$\Delta E_{ji} = \begin{pmatrix} R_j^T R_i & R_j^T (t_i - t_j) \\ 0 & 1 \end{pmatrix}. \tag{6}$$

While the orthogonality of $R_i$ and $R_j$ ensures that $R_j^T R_i = R_i^T R_j$, it is clear that $R_i^T(t_j - t_i) \neq R_j^T(t_i - t_j)$. Consequently, the relative translation angle—and hence AUC@30—is sensitive to the ordering of input images, which can lead to differences exceeding five points. To mitigate this, we include both $E_{ij}$ and $E_{ji}$ in the relative pose sequence instead of only $E_{ij}$. This modification preserves the relative rotation error while introducing permutation equivariance to relative translation error and AUC@30, resulting in a more robust and fair evaluation of pose estimation accuracy.

## D  LARGE LANGUAGE MODEL USAGE

We used LLMs solely as a writing assistant to improve grammar, clarity, and conciseness of the manuscript. The research ideas, technical contributions, experiments, and analyses were entirely conceived and conducted by the authors. No content was generated by LLMs beyond language refinement, and all scientific claims and results are the sole responsibility of the authors.

