# OpenReview forum: "VGGT-X: When VGGT Meets Dense Novel View Synthesis"
_ICLR.cc/2026/Conference — ICLR 2026 Conference Withdrawn Submission_

### Official Review · Reviewer_EmR9 · 2025-10-29

**Soundness:** 2
**Presentation:** 3
**Contribution:** 2
**Rating:** 4
**Confidence:** 3

**Summary:**

This paper explores extending 3D Foundation Models (3DFMs), particularly VGGT, to dense novel view synthesis (NVS), a regime involving hundreds or thousands of input views. The authors identify two critical bottlenecks in scaling 3DFMs: (1) high VRAM cost and (2) noisy predictions that degrade 3D Gaussian Splatting (3DGS) training.

They propose **VGGT-X**, an enhanced pipeline that includes:

- A memory-efficient implementation of VGGT (VGGT− and VGGT−−) using feature pruning, mixed precision (BFloat16), and chunked frame-wise processing to scale inference to 1000+ views.
- An adaptive global alignment (GA) module leveraging XFeat correspondences with an epipolar loss and adaptive weighting to refine camera poses.
- A robust 3DGS training strategy combining 3DGS-MCMC and joint pose optimization to handle imperfect initializations.

Experiments across datasets demonstrate state-of-the-art performance in both pose estimation and rendering quality for COLMAP-free pipelines. While some gaps remain versus COLMAP-initialized baselines, the paper provides diagnostic analyses and discusses overfitting and generalization challenges.

**Strengths:**

1. **Clear identification of key scalability issues** in current 3DFMs and dense NVS (memory inefficiency and initialization noise).
2. **Technically sound improvements**:
    - Efficient VGGT scaling to over 1000 images without accuracy loss.
    - Adaptive global alignment that effectively refines noisy camera poses with learned correspondence weighting and learning-rate adaptation.
3. **Comprehensive experiments** with ablations across multiple components (memory optimization, pose refinement, and training strategies) showing consistent gains over prior methods like 3RGS, CF-3DGS, and HT-3DGS.
4. **Insightful discussion** about residual overfitting and the generalization gap.
5. **Clarity and completeness**: Figures, tables, and methodological explanations are thorough and well-structured.

**Weaknesses:**

1. **Marginal novelty** in some components: Much of the contribution lies in **engineering optimization and integration**, not in introducing fundamentally new architectures.
2. Although training-set performance approaches COLMAP-initialized baselines, test-time quality lags, suggesting limited robustness. The paper asserts that COLMAP is often slow and fragile in low-texture or low-overlap captures, but provides no targeted evaluation to substantiate this claim. Experiments on deliberately **low-texture / low-overlap** scenarios, where 3DFMs should have an advantage, would clarify the robustness gap and more convincingly demonstrate VGGT-X’s benefits over COLMAP.
3. **Dependency on a specific matcher**: GA relies on **XFeat**; the ablation shows large drops when using VGGT’s tracking head. It’s unclear how robust results are to other lightweight matchers (e.g., SIFT+RANSAC on large scenes), or to degraded match quality (textureless/low-overlap).
4. **Comparisons and fairness quirks**: 3RGS is run on **the authors’ own VGGT−−+GA outputs** (Tab. 1–2). While motivated for fairness, this also means 3RGS is evaluated in a partially “ours” setting.
5. **Clarity/details**: Some specifics are buried: exact runtime of GA vs BA on long sequences; chunk size sensitivity (**S=128** is fixed); numerical stability of BF16 across GPUs.

**Questions:**

1. Does VGGT-X still operate when multi-view inputs are provided in an unordered, unconstrained manner (as in COLMAP’s default assumption)? In particular, how do the chunking and alignment stages behave when (a) images within a chunk have little or no overlap, or (b) the alignment step fails to find reliable matches? Do you have safeguards: e.g., adaptive re-grouping of chunks, cross-chunk matching, fallback strategies (tracking/BA), or discarding problematic chunks and can you report failure rates or thresholds that trigger these behaviors?
2. In Table 1 under the COLMAP-initialization–free setting, the Mip-NeRF360 metrics appear significantly lower than the values reported for 3DGS-MCMC. Where does this gap come from?
3. How sensitive is the proposed adaptive weighting function to its hyperparameter α and histogram binning choices?
4. How does the method perform under varying scene overlaps or low-texture regions—are certain environments particularly problematic?

---

### Official Review · Reviewer_9LLq · 2025-10-30

**Soundness:** 2
**Presentation:** 3
**Contribution:** 1
**Rating:** 2
**Confidence:** 3

**Summary:**

This paper presents VGGT-X, which extends upon VGGT for handling dense views. The methodology consists of twofolds, where it first reduces the computational complexity of VGGT by optimizing its code implementation, followed by training 3DGS upon the predictions from VGGT.

**Strengths:**

1. The paper tackles practical problem, where it aims to reconstruct 3D scene from dense image views in an efficient manner.

2. The proposed method achieves state-of-the-art performance among 3DGS methods without using COLMAP.

**Weaknesses:**

1. The method lacks novelty as it heavily relies on existing solutions for each of its components. Especially, MCMC is also introduced in 3RGS for the exact same reason, and the epipolar constraint is also commonly used in the literature.

2. The proposed VGGT--, which is a lightweight version of the original VGGT, is basically refactored version of VGGT for lowering the memory footprint. This would be considered as more of an implementation detail, rather than an academic contribution.

3. Overall, this makes the paper hard to be considered as an academic paper for ICLR. Not only the method does not learn "representations" at all, the reviewer at least expects the paper to study VGGT in a more in-depth manner to originate its limitations discussed in the paper. However, the paper mainly resorts to empirical results for justifying additional modules to "patch" VGGT, such as the usage of XFeat.

**Questions:**

1. Are any of the problems, such as inaccurate camera poses or correspondence, unique problem in the dense NVS setting? Consequently, are the techniques used in the paper especially effective to the dense setting, or can they be equally applied when having sparse views? It would be helpful to provide analysis on why the pose and the correspondence from VGGT is underwhelming, and must be solved with solutions involving external modules or heuristic methods.

---

### Official Review · Reviewer_5AJt · 2025-10-31

**Soundness:** 3
**Presentation:** 3
**Contribution:** 3
**Rating:** 6
**Confidence:** 5

**Summary:**

This paper addresses the key challenges of applying 3D Foundation Models (3DFMs) to dense Novel View Synthesis (NVS), namely the prohibitive VRAM consumption and the noisy initial predictions that degrade reconstruction quality. To overcome these barriers, the authors introduce VGGT-X, a comprehensive framework that integrates a memory-efficient VGGT implementation capable of handling over 1,000 images, an adaptive global alignment strategy to refine camera poses, and a robust 3D Gaussian Splatting (3DGS) training method tailored for imperfect initializations. Experiments demonstrate that this approach achieves state-of-the-art performance in both pose estimation and rendering quality for COLMAP-free NVS, substantially narrowing the performance gap compared to methods reliant on traditional Structure-from-Motion pipelines and representing a significant step towards building fast, reliable, and fully automated dense NVS systems.

**Strengths:**

Originality: The paper's main innovation is not a single novel algorithm, but a creative and effective integration of several techniques to solve a well-defined problem. It systematically adapts and enhances an existing 3D Foundation Model (VGGT) to work in a dense-view regime where it previously failed, which constitutes a novel and practical contribution.

Quality: The technical quality is high, supported by extensive experiments and comprehensive ablation studies across multiple challenging datasets. The authors convincingly demonstrate the effectiveness of each component of their proposed VGGT-X framework, achieving state-of-the-art results for COLMAP-free methods.

Clarity: The paper is exceptionally well-written and easy to follow. It clearly articulates the core challenges of scaling 3DFMs, logically presents its multi-part solution, and uses figures and tables effectively to support its claims and illustrate the proposed pipeline.

Significance: This work is highly significant as it addresses a major bottleneck in 3D computer vision—the reliance on slow Structure-from-Motion (SfM) pipelines like COLMAP. By providing a practical and effective pathway to a fully COLMAP-free system for dense NVS, the paper has the potential to accelerate research and applications in automated 3D reconstruction and rendering.

**Weaknesses:**

W1: The overfitting issue is observed but not fully analyzed; it's unclear if the degraded test performance stems from flawed 3D geometry or overfit appearance features such as spherical harmonics.

W2: The paper notes a weaker performance on the CO3Dv2 dataset, but a deeper explanation is missing; there is no detailed qualitative analysis of how characteristics like object-centric scenes, sparse textures, or camera motion might specifically affect the model's predictions.

W3: Several key hyperparameters, particularly those governing the adaptive weighting and learning rate schedule, are presented without an ablation or sensitivity analysis, which makes it difficult to assess their robustness and generalizability across different datasets.

W4: The paper lacks comparative experiments with other methods [1-3] on scenarios with fewer input views, which would be crucial for understanding its performance in varying data availability conditions.

[1] NeRF-mm: Neural Radiance Fields Without Known Camera Parameters
[2] NoPe-NeRF: Optimising Neural Radiance Field with No Pose Prior
[3] InstantSplat: Sparse-view SfM-free Gaussian Splatting in Seconds

**Questions:**

See Weakness.

---

### Official Review · Reviewer_ddzG · 2025-11-01

**Soundness:** 3
**Presentation:** 3
**Contribution:** 2
**Rating:** 2
**Confidence:** 4

**Summary:**

This paper proposes VGGT-X, a pipeline to train 3DGS given a set of dense images without known camera parameters. Instead of following the traditional pipeline of running structure-from-motion (SfM) algorithms such as COLMAP to obtain camera poses which often provide noisy or sparse point clouds and are typically slow, the proposed method leverages recent advances in feed-forward geometry estimation using VGGT. To enable VGGT to process large number of images, the paper makes several changes to the forward pass such as discard unnecessary features (VGGT-) and reducing precision and adopting chunk wise processing in the frame-wise attention stage (VGGT—). Given the initial estimation of VGGT—, the paper proposes a global alignment step based on the corresponding points between views obtained from a off-the-shelf correspondence network XFeat. From the refined camera poses, they leverage 3DGS-MCMC to train 3DGS for scene reconstruction, which achieves lower but comparable results with 3DGS trained with camera poses obtained from COLMAP.

**Strengths:**

- The tackled problem of applying recent advances of 3D geometry foundation models to dense 3D novel view synthesis is an interesting direction.
- The overall writing is clear and thorough, making it easy to understand most of the proposed methods.
- The motivation and implementation to make VGGT’s inference more efficient is intuitive and straightforward.
- The proposed global alignment step with correspondence information from off-the-shelf correspondence network makes the alignment step to be done much quicker than the bundle adjustment done in previous works.
- Leveraging 3DGS-MCMC with pose optimization is also straightforward.

**Weaknesses:**

- **Lack of comparison with COLMAP:** As mentioned in the abstract and introduction section of the paper, this work starts by pointing out several problems of the original NVS pipeline depending on COLMAP, which is slow and struggles in low-texture or low-overlap captures. However, there is no direct comparison of the computation time and overall training time of the proposed VGGT-X and traditional COLMAP + 3DGS or COLMAP + 3DGS-MCMC, which makes it difficult to understand the advantages of utilizing the proposed method instead of COLMAP. In addition, as the current evaluation datasets (MipNeRF360, Tanks and Temples, and CO3D) is not typically low-texture datasets, it would be better to compare VGGT-X with COLMAP + 3DGS or COLMAP + 3DGS-MCMC in these scenarios, I would recommend looking into scenes in RealEstate10K[1], as some of the scenes in the dataset contain images with large textureless regions.
- **Unclear optimization strategy during global optimization step:** During global optimization, the method optimizes camera poses based on the correspondences obtained from a off-the-shelf model XFeat. As this process is disjoint with the VGGT estimation process, when optimizing the camera poses, how is the pointmaps from each view correctly modified?
- **Heuristic parameters:** Leveraging different learning rates by evaluating the quality of VGGT’s estimation through the epipolar distance is highly heuristic, which would require the users to re-explore adequate learning rates when applying in a new scene with different scales.
- **Consideration of more complicated noise:** The proposed method seem to mainly consider noise only in the camera extrinsic parameters, without further consideration of noise in the estimated camera intrinsics or other distortions as considered in SC-NeRF [2].

Overall, although the proposed method is straightforward, I am not convinced about the contribution of this work to the research community. The adoption of 3D foundation models like VGGT for dense-view NVS is interesting, but the main bottleneck seems to be the computation, and the proposed method of reducing the overall computation (VGGT -> VGGT --) seems to be more of an engineering effort rather than revealing new insights. In addition, the global optimization pipeline also contains heuristically defined parameters with large dependence on off-the-shelf correspondence networks, making the pipeline prone to error accumulation in both steps. The comparison with the original SfM algorithm COLMAP is also limited, making it hard to understand the advantages of the proposed method. As a result, at the point of submission, this work does not seem to meet the bar of ICLR.

### References

---

[1] Zhou, Tinghui, et al. "Stereo magnification: Learning view synthesis using multiplane images." arXiv preprint arXiv:1805.09817 (2018).

[2] Jeong, Yoonwoo, et al. "Self-calibrating neural radiance fields." Proceedings of the IEEE/CVF international conference on computer vision. 2021.

**Questions:**

Most of my concerns are listed in the weakness section. One addition question is listed below:

Q1. How does leveraging an off-the-shelf correspondence network increase robustness in low-textureless regions?

---

### Note · Authors · 2025-11-14

**Comment:**

We sincerely appreciate the reviewers for their constructive feedback and the valuable time they invested in evaluating our submission. Their insights are highly appreciated and will guide our future improvements. Thank you for the thoughtful review process.

**Withdrawal Confirmation:**

I have read and agree with the venue's withdrawal policy on behalf of myself and my co-authors.